# Liquid Biopsy as a Diagnostic and Prognostic Tool for Women and Female Dogs with Breast Cancer

**DOI:** 10.3390/cancers13205233

**Published:** 2021-10-19

**Authors:** Jucimara Colombo, Marina Gobbe Moschetta-Pinheiro, Adriana Alonso Novais, Bruna Ribeiro Stoppe, Enrico Dumbra Bonini, Francine Moraes Gonçalves, Heidge Fukumasu, Luiz Lehmann Coutinho, Luiz Gustavo de Almeida Chuffa, Debora Aparecida Pires de Campos Zuccari

**Affiliations:** 1Laboratory of Molecular Investigation in Cancer (LIMC), Department of Molecular Biology, Faculdade de Medicina de São José, São José do Rio Preto 15090-000, Brazil; jucimara.colombo@famerp.br (J.C.); marina.moschetta@docente.unip.br (M.G.M.-P.); adriana.novais@ufmt.br (A.A.N.); bruna.stoppe@edu.famerp.br (B.R.S.); enrico.bonini@edu.famerp.br (E.D.B.); francine.goncalves@unifipa.com.br (F.M.G.); 2Laboratory of Comparative and Translational Oncology (LOCT), Department of Veterinary Medicine, Faculty of Animal Science and Food Engineering, University of Sao Paulo, Pirassununga 13635-900, Brazil; fukumasu@usp.br; 3Luiz de Queiroz College of Agriculture (ESALQ), University of São Paulo, Piracicaba 13418-900, Brazil; llcoutinho@usp.br; 4Department of Structural and Functional Biology, Institute of Biosciences of Botucatu, Universidade Estadual Paulista, Botucatu 18618-689, Brazil; luiz-gustavo.chuffa@unesp.br

**Keywords:** breast cancer, circulating tumor DNA (ctDNA), liquid biopsy, next generation sequencing (NGS), gene variant

## Abstract

**Simple Summary:**

Breast cancer (BC) has common characteristics in women and female dogs, such as high recurrence, metastasis, and mortality rate. In both species, BC prognosis is limited due to its heterogeneous molecular aspects. Although conventional biopsy remains the gold standard for BC diagnosis, liquid biopsy is a very promising tool, especially for patient follow-up. We investigated the effectiveness of liquid biopsy in the diagnosis and follow-up of women and female dogs with BC, using both core biopsy and plasma samples processed by next generation sequencing (NGS) assay. We noted that NGS is a sophisticated technique generating multiple and complex results, which must be validated. Notably, the number of genetic variants increased as the disease progressed. We conclude that liquid biopsy can be considered more effective when performed from the onset of the disease and continues to be applied for monitoring the follow up of BC patients, helping to drive the clinician’s decision for medical intervention.

**Abstract:**

Introduction: Breast cancer (BC) is the malignant neoplasm with the highest mortality rate in women and female dogs are good models to study BC. Objective: We investigated the efficacy of liquid biopsy to detect gene mutations in the diagnosis and follow-up of women and female dogs with BC. Materials and Methods: In this study, 57 and 37 BC samples were collected from women and female dogs, respectively. After core biopsy and plasma samples were collected, the DNA and ctDNA of the tumor fragments and plasma were processed for next generation sequencing (NGS) assay. After preprocessing of the data, they were submitted to the Genome Analysis ToolKit (GATK). Results: In women, 1788 variants were identified in tumor fragments and 221 variants in plasma; 66 variants were simultaneously detected in tumors and plasma. Conversely, in female dogs, 1430 variants were found in plasma and 695 variants in tumor fragments; 59 variants were simultaneously identified in tumors and plasma. The most frequently mutated genes in the tumor fragments of women were *USH2A*, *ATM*, and *IGF2R*; in female dogs, they were *USH2A*, *BRCA2*, and *RRM2*. Plasma of women showed the most frequent genetic variations in the MAP3K1, BRCA1, and GRB7 genes, whereas plasma from female dogs had variations in the *NF1*, *ERBB2*, and *KRT17* genes. Mutations in the *AKT1*, *PIK3CA*, and *BRIP* genes were associated with tumor recurrence, with a highly pathogenic variant in *PIK3CA* being particularly prominent. We also detected a gain-of-function mutation in the *GRB7*, *MAP3K1*, and *MLH1* genes. Conclusion: Liquid biopsy is useful to identify specific genetic variations at the beginning of BC manifestation and may be accompanied over the entire follow-up period, thereby supporting the clinicians in refining interventions.

## 1. Introduction

Breast cancer (BC) in women and dogs shares similarities, such as high rate of recurrence and metastasis and a high mortality rate [1,2]. BC represents the second leading cause of cancer death in women worldwide [3]. In both species, the prognosis of the disease is often limited since it is a very heterogeneous type of neoplasia, characterized by the coexistence of different cell clones in the tumor. Such heterogeneity is considered the essential driving force for tumor clonal evolution, which can be induced by chemotherapy as a result of the expansion of resistant cell clones; these cell clones are the main cause of tumor recurrence and repeated failures in cancer treatment [4,5].

BC is a complex disease encompassing several distinct entities at the molecular and clinical presentation. Hereditary BC is represented mainly by mutations in the *BRCA1* and *BRCA2* genes [6,7] and accounts for only a small percentage of cases; otherwise, most BCs are sporadic and result from the accumulation of somatic changes [8]. Recent advances in sequencing-based technologies have provided understanding of genetic changes and deregulation of oncogenic signaling pathways, involving growth signaling, stress-related response, metabolism, and cell–cell communication, which affect the growth and progression of cancer. Together with the host’s response to cancer, these somatic changes determine the clinical course of the disease [9,10].

Although conventional biopsy remains the gold standard for the diagnosis of BC [11], liquid biopsy is a very promising tool, especially for patient monitoring. Genetic diversity and dynamic changes in genomic profiles can be determined and accompanied through liquid biopsy, which allows for a better precision of prognosis and treatment [12].

Therefore, using circulating tumor DNA (ctDNA), it is possible to detect genetic alterations in the tumor, such as specific mutations that arise from the disease and during therapy, both in human and canine BC. Somatic mutations in *AKT1*, *PIK3CA*, *PTEN* and *TP53* genes are found at high frequency in human BC, representing 26.4% in *PIK3CA*, 24.7% in *TP53*, 3.8% in *PTEN,* and 2.8% in *AKT1*, according to the Somatic Mutations in Cancer Catalog (COSMIC) [13]. These changes are indicative of progression or may be responsible for the chemoresistance and consequent recurrence and spread of tumor, resulting from the process of clonal evolution [14,15].

There is a growing need to monitor the genomic profiles of tumor cells at the beginning of the process and in the patient follow-up, especially to detect disease recurrence and metastasis. However, repeat tissue biopsy is not practicable, whereas circulating tumor cells detached from a primary tumor are present in the bloodstream and can be easily obtained [16,17]. Liquid biopsy using ctDNA is a non-invasive and replicable method, being useful for tumor cell counting, pathological characterization and molecular assay. In addition, liquid biopsy with ctDNA may replace tissue biopsy to predict drug sensitivity and resistance, monitor drug responsiveness, and to examine metastasis [18,19,20].

In this context, one of the techniques of special interest is next-generation sequencing (NGS), which has been incorporated into clinical practice to identify mutations in cancer patients while targeting treatment with specific drugs [21,22]. The advent of NGS panels in clinical practice favors novel therapeutic choices, particularly for patients with limited therapy options [23].

Therefore, the objective of this work was to verify the effectiveness of liquid biopsy in detecting variants of interest using plasma and tumor fragments obtained during the diagnosis and also during the monitoring of women and female dogs with BC using NGS technique.

## 2. Materials and Methods

### 2.1. Ethics

The study was approved by the Research Ethics Committee (CEP) (Protocol number CAAE 83446118.5.00005415) and by the Ethics Committee on the Use of Animals (CEUA) (Protocol number 001-003244/2013) of the Faculdade de Medicina de São José do Rio Preto, (FAMERP).

### 2.2. Sample Collection

Women attended and were diagnosed with BC at the Gynecology and Obstetrics Section of the Medical School of São José do Rio Preto, (FAMERP). A total of 57 samples were analyzed: plasma (*n* = 16) and core biopsy (*n* = 16) from women newly diagnosed with BC and plasma (*n* = 4) from women, comprising four relapses and 11 remissions; 10 control women without previous disease were also evaluated. The inclusion criterion was the presence of a malignant breast tumor, confirmed by histopathological examination. The exclusion criterion for the control group was the presence or history of cancer. All information regarding the prognosis (response to therapy, development of metastases, recurrences or deaths), and results of anatomopathological and immunohistochemical tests of patients were obtained through consultation in the electronic medical record of the patients.

For canine BC, 37 samples were examined: 11 from tumor fragments and plasma of female dogs with newly diagnosed BC. We also studied eight follow-up dogs (mastectomized) and seven dogs without neoplasms obtained from the Veterinary Clinics in São José do Rio Preto (SP), Jaboticabal (SP), Catanduva (SP), São Paulo (SP), and Sinop (MT), Brazil. The inclusion criteria for the canine species were: presence of breast tumor(s) with or without metastasis and absence of comorbidities. For the control group, the exclusion criteria were: presence or history of neoplasia. Blood and fragment collections for the newly diagnosed group were performed at the time of surgical excision and blood collection from the control group and the follow-up group was carried out in routine and return visits, respectively. In the canine species, information regarding the prognosis (response to therapy, development of metastases, relapses or deaths) were obtained from clinical form.

### 2.3. DNA and ctDNA Extraction

The DNA and ctDNA of tumor fragments and plasma from women and female dogs were extracted using the AllPrep^®^ DNA/RNA/Protein Mini Kit (Qiagen, Hilden, Germany) and QIAmp Circulating Nucleic Acid Kit (Qiagen^®^), respectively. The integrity and quality of the extracted DNAs were verified by the Qubit Fluorometric Quantitation equipment (Thermo Fisher Scientific, Santa Clara, CA, USA).

### 2.4. Preparation of DNA and ctDNA Sequencing Libraries for Illumina System

The genomic libraries were built with the SureSelectXT Low Input Target Enrichment System for Illumina Paired-End Multiplexed Sequencing Library Kit (Agilent Technologies, Santa Clara, CA, USA), following the manufacturer’s recommendations. After DNA and ctDNA were fragmented, and end repair and 3’ends adenylation of fragments was performed by adding a nucleotide (a-tailing Mix) to the 3´end in order to prevent them from binding to each other during the ligation of adapters. Bar-coded adapters were ligated to the DNA fragments and a PCR reaction was performed to produce the sequencing libraries. The quality of libraries and quantification were performed using Agilent 2100 Bioanalyser and qPCR with KAPA Library Quantification kit (KAPA Biosystems, Foster City, CA, USA).

### 2.5. Next-Generation Sequencing

A 20 pM sample pool was loaded into specific cartridges for sequencing on the Illumina MiSeq^®^ with the Illumina^®^ MiSeq Reagent Kits v2 (300 cycles) (Illumina, San Diego, CA, USA). The kit generates around 4.5–5 Gb and running time was approximately 24 h. The sequencing data were extracted in fastQC format, making it possible to check the quality metrics. The samples were distributed in five cartridges. A panel of 168 genes involved in breast carcinogenesis were analyzed (Appendix A), being 89 from humans and 79 from dogs. The Illumina Experiment Manager^®^ program (Illumina, San Diego, CA, USA) was oriented to associate each identified read. Vertical and horizontal sequencing coverage was 200 times for DNA samples extracted from tumor fragments and 2000 times for samples extracted from free circulating plasma DNA. This criterion was determined since tumor samples theoretically have higher quality and quantity than free DNA samples circulating in plasma, allowing the number of base readings to be lower.

### 2.6. Bioinformatics Analysis

The quality values of the sequences were obtained using FastQC. After pre-processing the results, in BAM format, data were subjected to workflow, according to the good practices of GATK (Genome Analysis ToolKit, from the Broad Institute, USA) (Figure 1). The hg38 version of human genome and the CanFam3.1 version of canine genome were used as references for all data processing of women and dogs, respectively. In both species, a pool of normal breast samples (PoN) was used to filter out non-tumor variants. using the tools Mutect2, GenomicsDBImport, and CreateSomaticPanelOfNormals from the GTAK package. For the women’s samples, the vcf file of the genomAD project (germline variants) was used to reduce the contamination of germline variants. For the dog’s samples, a normal panel of germline variants (https://data.broadinstitute.org/vgb/dog/dog/canFam3/variation/broad.canine.pon.germline.snps.vcf.gz; accessed on 20 May 2021) was also used to reduce contamination of these variants. Only variants which passed through the GATK filters, that is, the somatic variants, were included. The annotation of women and dogs was performed using the Annovar and VEP programs, respectively.

The output from the previous step (filtered vcf file) was used for annotation, using the Annovar program, in the case of women, and the VEP program (from Ensembl) in the case of female dogs.

### 2.7. Statistical Analysis

The results were analyzed by R test (version 3). The association between the clinical outcome of women and the occurrence of mutations was evaluated using Fisher’s exact test. Statistical significance was set at *p* < 0.05. Graphics were generated from GATK (Genome Analysis Toolkit, Broad Institute, Cambridge, MA, EUA).

## 3. Results

### 3.1. Characterizing Human and Canine Population with BC

Considering the newly diagnosed group, the mean age was 58.18 years, with the majority of women displaying invasive ductal carcinoma (26.66% grade 1 and 73.33% grade 2). The molecular subtype was based on immunohistochemical tests (12.50% luminal subtype A, 31.25% luminal B, 12.50% hybrid luminal, and 37.50% triple negative. The sample P12/C12 was diagnosed as fibroadenoma; however, it was a borderline tumor. The patient had high Ki-67 and underwent radiotherapy. Therefore, despite being diagnosed as fibroadenoma, the tumor had an aggressive behavior and showed important genetic variants in the NGS analysis (Table 1). The mean age of the follow-up group (disease recurrence) was 52.25 years (50% had the luminal molecular subtype B, 25% had HER2 positive and 25% had triple negative) (Table 2). Regarding the patients in remission stage, the mean age was 60.90, and 1 patient (9.09%) was in remission for 4 months, two patients (18.18%) for 3 years, and eight patients (72.72%) for more than five years. Among the treatments used in the remission group, three patients (27.27%) underwent chemotherapy combined with radiotherapy, three (27.27%) received only chemotherapy, 1 (9.09%) received only radiotherapy, one (9.09%) received hormone therapy, one (9.09%) underwent only mastectomy, one (9.09%) underwent mastectomy followed by chemotherapy and radiotherapy, and one (9.09%) was treated with radiotherapy combined with hormone therapy (Table 3). The mean age of the control group was 59.50 years (Table 4).

The age of newly diagnosed female dogs ranged from 7 to 16 years, with an average age of 11.72 years. Due to the miscegenation of the population, most animals were SRD, i.e., without a defined race. Based on the histopathological information, three patients were classified as T1, 5 as T2, and 3 as T3; 10 as N0 and 1 as N1; 1 as M1. The histopathological features of breast tumors from 11 female dogs were multiple: represented by three mixed carcinomas, grade I; two complex carcinomas, grade II; one tubular carcinoma, grade I; one breast osteoma, one breast osteosarcoma, one ductal apocrine carcinoma, one complex adenoma, and one mast cell tumor (Table 5). It is noteworthy that ductal apocrine and mast cell carcinomas were initially diagnosed as breast cancer, but later, they were characterized as skin cancers. They were maintained in the study due to its high frequency of mutations. The age female dogs in follow-up ranged from 8 to 13 years, with an average age of 10.62 years. Based on the histopathological information, four patients were classified as T2, 2 as T3; 2 as N0, 3 as N1 and 1 as N2 as N1; 4 as M1. The histopathological features of breast tumors from 8 female dogs were: one papillary carcinoma, grade II; one tubular carcinoma, grade III; one carcinosarcoma, one mixed carcinoma, grade II; one osteosarcoma; and one complex carcinoma, grade I (Table 6).

### 3.2. Detection of Individual Relevant Genetic Variants in Tumor Fragments and in Plasma of Women with BC

To examine the efficacy and functionality of liquid biopsy, high quality samples were extracted from plasma and tumor fragments to generate genomic library construction. In women, a total of 1788 and 221 gene variants were detected in tumor fragments and plasma, respectively; 66 gene variants were simultaneously identified in patients’ tumors and plasma. Among these variants, 24 were located in the exon regions (exonic variants) (Appendix A).

The most commonly mutated variant found in women fragments was the *missense* variant. The main single nucleotide variant in tumor fragments of women was C > A (57.49%) and C > T (29.73%) (Figure 2A,B). The most frequent top mutated genes in tumor fragments of women were *USH2A, ATM, IGF2R, MKI67,* and *MAP3K1* (median variants per sample = 20; missense, nonsense, and splice site mutations) (Figure 2C,D).

After examining the plasma of BC patients, we observed that missense variation was the most common or abundant mutated variant in women. The main single-nucleotide variant in plasma of women was C > T (53.65%) and C > A (17.07%) (Figure 3A,B). The most frequent top mutated gene in the plasma of women were *MAP3K1, BRCA1, GRB7, USH2A*, and *TP53* (median variants per sample = 2; missense, inframe deletion, and frameshift mutations) (Figure 3C,D).

### 3.3. Detection of Individual Relevant Genetic Variants in Tumor Fragments and in Plasma of Dogs with BC

In female dogs, a total of 695 and 1430 gene variants were detected in tumor fragments and plasma, respectively; 59 gene variants were simultaneously identified in female dogs’ tumors and plasma (Appendix A).

The most commonly mutated variant found in female dog fragments was the *missense* variant. The predominant nucleotide variants in female dogs was T > C (23.89%) and C > T (20.68%) (Figure 4A,B). The most frequent top mutated genes in female dogs, *USH2A, BRCA2* and *RM2, CEP55*, and *GATA3* genes were the most representative mutated genes (median variants per sample = 31; missense, nonsense, splice site, and frameshift mutations) (Figure 4C,D).

After examining the plasma of BC patients, we observed that *missense* variation was the most common or abundant mutated variant in canine samples. The main single nucleotide variant in plasma of female dogs was C > T (29.27%) and T > C (26.42%) (Figure 5A,B). The most frequent top mutated genes in the plasma of women were *MAP3K1, BRCA1, GRB7, USH2A*, and *TP53* (median variants per sample = 2; missense, inframe deletion, and frameshift mutations). In female dogs, *NF1, ERBB2, KRT17, PMS2*, and *MAP3K1* genes were the most representative mutated genes (median variants per sample = 56; missense and frameshift mutations) (Figure 5C,D).

### 3.4. Breast Cancer Driver Genes in Women Share Similar Variant Mutations in Tumor Fragments and Plasma

We sequenced 89 BC-related genes in 15 newly diagnosed breast cancer patients. Among the sequenced gene mutations, 40 genes exhibited at least one driver somatic mutation that included missense (64.18%), nonsense (8.89%), inframe (0.44%), frameshift (0.66%), synonymous (25.58%), startloss (0.11%) and stoploss (0.11%) mutations. The five most affected genes were *MAP3K1* (altered in 10 samples, 62%), followed by *USH2A* (10 cases, 62%), *ATM* (9 cases, 56%), *IGF2R* (9 cases, 56%), and *EGFR* (8 cases, 50%) (Figure 6A). Moreover, 89 genes were sequenced in plasma samples of 16 newly diagnosed breast cancer patients, and 18 genes presented missense (53.19%), inframe (19.14%), frameshift (6.38%) and synonymous (21.27%) mutations. The most affected genes were *MAP3K1* (altered in eight samples, 67%), *BRCA1*, *ERBB2*, *FOXC1*, and *GRB7* (2 cases, 17%) (Figure 6B).

Considering the common variants shared by tumor fragments and plasma of women, we identified important gene variations in *ACTR3B* (C > T mutation), *BRCA1* (T > A), *CDC6* (C > T), *CENPF* (T > A), *CHEK2* (T > C), *EXO1, GATA3*, and *GRB7* (G > A), *IGF2R* (G > A and T > C), *KIF2C* (G > A), *KRT5* (G > T and A > TG), *MAP3K1* (C > T and CAA > -), *MKI67* (T > C), *MMP11* (T > C), *MYBL2* (C > T), *PMS2* (C > T), *TP53* (T > C), *TYMS* (T > C), *USH2A* (G > A, G > T, and T > G) (Figure 7A). Notably, while the somatic mutation rate of *BRCA1* was 43.7%, the mutation rate of *TP53* was 50% (Figure 7B).

We also performed the analysis of genetic variants in the plasma of women with BC who are in remission or with recurrence. Thus, in women in remission, we observed missense (25.14%), nonsense (0.28%), inframe (0.57%), frameshift (0.57%) and synonymous (73.40%) mutations. On the other hand, women with recurrent disease presented missense (85.71%) and inframe (14.28%) mutations. We further identified 10 variants coexisting between patients in remission and after relapsed disease which were detected in *AKT1*, *NF1*, *AURKA, MSH6, MAP3K1*, *ALOX12-A, MTOR, TYMS*, *CDH1*, and *DLG2* genes (Figure 7C). Moreover, in women, there was also an association between mutations in *AKT1* (CA > TG, *p* = 0.01), *PIK3CA* (A > G, *p* = 0.02) and *BRIP* (T > C, *p* = 0.02) genes with tumor recurrence, highlighting a highly pathogenic variant in *PIK3CA* gene, according to CLINVAR. We observed that women in remission had more variants than women with recurrence. This probably occurred because some patients classified within the remission group did not show, at the time of sample collection, clinical evolution of disease; however, the disease progressed during the course of the study.

### 3.5. Breast Cancer Driver Genes in Dogs Share Similar Variant Mutations in Tumor Fragments and Plasma

Regarding the female dogs, we sequenced 79 BC-related genes, and those with high number of variants shared by tumor fragments and plasma were *GATA3* and *mTOR* (missense), *SFRP1* (frameshift insertion), *BRCA2* (multi hit), *FOXC2, ATM, TGFBR3,* and *BRIP1* (missense and multi hit mutations). A total of 56 genes exhibited at least one somatic mutation in tumor fragments including missense (76.83%), nonsense (1.87%), inframe (5.88%), frameshift (10.92%) and splice (4.46%) mutations. The most affected genes were *BRCA2* (altered in 13 samples, inframe and multi hit mutations), *CEP55* (13 cases, frameshift, missense, and multi hit mutations), *GATA3* (13 cases, missense mutation), *SFRP1* (13 cases, frameshift and multi hit mutations), and *USH2A* (13 cases, missense and multi hit mutations) (Figure 8A). Additionally, from 79 genes sequenced in plasma samples of dogs with BC, we identified 61 genes harboring missense (87.27%), nonsense (1.88%), inframe (1.87%), frameshift (7.47%), and splice (1.46%) mutations. The most affected genes were *GATA3* (altered in 11 samples, missense mutation), *PMS2* (10 cases, missense and multi hit mutations), *KRT17* (9 cases, missense and multi hit mutations), *MAP3K1* (9 cases, missense, splice, and multi hit mutations), and *NF1* (9 cases, missense and multi hit mutations) (Figure 8B).

Considering the most representative variants shared by tumor fragments and plasma of dogs, we identified important gene variations in *ATM* (C > A mutation), *CCNE1* (G > C), *FGFR4* (C > T), and *GATA3* (G > A and C > CT) (Figure 9). The genetic variants in the plasma of dogs with BC were evaluated during the follow-up. The most common mutations were missense (75.37%), inframe (12.67%), frameshift (7.46%) and splice (4.47%). In dogs, the G > C variant (missense mutation) of *CCNE1* gene was detected in all paired samples of fragments and plasma (100.00%). In addition, the G > A and C > CT variants (missense mutation) of *GATA3* gene were commonly found in nine (81.8%) and eight (72.7%) paired fragment and plasma samples, respectively (Figure 9A). Figure 9B highlights the correlations between gene variants and female dogs’ samples. We further explored blood samples of dogs to identify gene variants during the patient follow-up. Of note, missense mutation in *AKT1, ATM, CDK2, GATA3,* and *ERBB2* genes, and inframe deletion in *mTOR* gene were commonly detected in both disease remission and metastasized disease (Figure 10), thereby revealing potential targets to be screened during the course of disease.

### 3.6. Main Variants Associated with Disease Progression Shared by Both Species

We found that the number of variants increased with disease progression. Women classified in the remission group (Rem5, Rem6, Rem7, Rem8 and Rem9) did not show clinical development of the disease at the time of sample collection. However, liquid biopsy analysis revealed a high number of mutations in these patients. This observation was consistent with the subsequent clinical evolution, as in these patients the disease progressed with the appearance of metastases.

After comparing the data between women and female dogs, both in the fragment and plasma samples, we observed important genes which were commonly altered. In tumor fragments, the *BRCA2* and *USH2A* genes showed many variants in both species. Otherwise, in the plasma, we detected several variants in the *ATM*, *ERBB2* and *MAP3K1* genes shared between women and female dogs.

## 4. Discussion

We performed genomic analysis in 57 BC samples in women, using *core biopsy* of fragments and plasma. By recognizing the similarities of BC molecular background in human and canine species, we further evaluated 37 BC samples (biopsy of fragments and plasma) in dogs. The canine mammary tumor (CMT) has been proposed as a promising model resembling woman BC, since they share common components regarding the physiopathology of the disease [2,24]. We extracted DNA from these samples to construct a genomic library for NGS, ensuring the high quality of the procedures with respect to the efficacy and functionality of liquid biopsy. While women had more mutations in the fragments vs. plasma, the dogs showed more mutations in the plasma vs. tumor fragments; this fact could be explained by individual tumor heterogeneity. The mammary tumors of female dogs are generally bulkier and more numerous than those of women. Therefore, fragments removed from female dogs will not represent all existing tumor clones. On the contrary, since BC is smaller in women, the fragment achieves a greater representation of the tumor as a whole [2,25].

Women with BC classified into the remission group had a high number of mutations. This fact was consistent with their later clinical evolution, as these patients presented disease progression with the occurrence of metastasis. This supports the importance of liquid biopsy in detecting mutations that can predict the clinical course of disease. The investigation of tumor variants in ctDNA, isolated from plasma, may provide substantial information regarding the tumor, such as the development of chemoresistance and presence of residual and recurrent disease [19,26].

These events occur because the genomic characteristics of a metastatic tumor are different from those found in a primary tumor, due to the interval between occurrences. Furthermore, genomic differences are intensified after treatment, including chemotherapy [27]. In order to monitor the mutational dynamics during a treatment, the acquisition of tissue biopsies would be inadequate and difficult to perform, since they are invasive procedures with potential clinical complications. Thus, the assessment of ctDNA is indisputably an ideal method, allowing the monitoring of response to treatment by evaluating the dynamics of genetic changes that occur in the tumor during exposure, for example, to a chemotherapeutic agent, being excellent for anticipating the recognition of relapse. In addition, conventional biopsy has shown limitations in terms of tumor representation, since neoplastic cells are heterogeneous [12,16].

Our data showed that most BC analyzed was classified as luminal type and missense mutations were the most frequent. In fact, this is in agreement with previous studies which documented that primary BC had more missense mutations in luminal/ER+ and epidermal growth receptor 2 positive (HER2+) subtypes while in triple negative breast cancers (TNBCs), nonsense, frameshift, and complex mutations were more common variations [8].

The *USH2A* gene was one of the most frequently mutated genes observed in tumor fragments from women and dogs, as well as in the plasma of women. This gene encodes the Usherin protein, which contains laminin EGF motifs, a pentraxin domain and several type III fibronectin motifs [28]. The mutation spectrum is very heterogeneous and includes over 1500 mutations with more than 690 variants presumed to be pathogenic, which span the whole USH2A gene, consisting of nonsense, missense, deletions, duplications, splicing variants and pseudo-exon inclusion variants [29].

Li et al. [30] detected *USH2A* variants in families that did not carry germline mutations of the *BRCA1* and *BRCA2* genes. They further concluded that these variants did not increase the risk of BC. In turn, Natraja et al. [31] verified the expression of chimeric transcripts in a chromosomal region comprising the *USH2A* gene in samples of micropapillary breast carcinomas. Furthermore, Santarpia et al. [8] also found variants in the USH2 gene in women with triple negative BC, which has a more reserved prognosis. In a recent work by Kim et al. [32], pathogenic germline variants of this gene were found in patients with BC. This supports the need for further studies that correlate variants of *USH2A* gene with BC risks.

In women, the *ACTR3B* gene showed concordance and a high frequency of somatic mutation in most tumor/plasma pairs. This gene is related to cytoskeleton and cell motility [33]. However, to date, there are no studies correlating *ACTR3B* changes with BC. The *AURKA* gene also showed concordance and high frequency of somatic mutation in most tumor/plasma pairs in women being further associated with tumor recurrence. This proto-oncogene encodes a protein kinase that plays a central role in mitosis and its overexpression or amplification was observed in several types of tumors, including BC. *AURKA* overexpression has been associated with a more aggressive phenotype and worse prognosis in BC patients [34].

The *CCNE1* and *GATA3* genes harbored somatic variants in most of the fragment/plasma pairs in dogs. The *CCNE1* gene encodes cyclin E, a key kinase complex for cell cycle regulation from G1 to S phase. *CCNE* gene amplification is highly associated with the development of BC, especially TNBC [35]. According to Huang et al. [7], *CCNE1* amplification represents an independent risk factor in non-*BRCA* carriers with TNBC. Moreover, Zhao et al. [35] reported that *CCNE1* amplification may confer resistance to chemotherapy and is associated with poor overall survival in patients with TNBC.

*GATA3* has emerged as a prominent transcription factor required for maintaining mammary gland homeostasis, and its loss is associated with aggressive BC development. Yu et al. [36] reported that *GATA3* and *UTX* expressions are positively correlated and demonstrated that GATA3/UTX complex synergistically regulates a cohort of genes including *DICER* and *UTX*, which are critically involved in the epithelial-to-mesenchymal transition (EMT). The *GATA3* gene encodes a transcription factor and is mutated in about 10% of BC. Recent genomic analysis of human BCs revealed high-frequency mutation in *GATA3* in luminal tumors, suggesting an important driver function [37]. Notably, the alterations found in tumor and plasma of women with BC were also observed in the work of D’Amico et al. [38]. According to the authors, the apparent lack of agreement between some specific alterations is potentially determined by intratumoral heterogeneity and can be overcome by plasma-based analysis of variants.

In the present study, we detected a pathogenic variant in the *PIK3CA* gene correlated with tumor recurrence. According to the Catalog of Somatic Mutations in Cancer (COSMIC), somatic mutations in the *PIK3CA* gene occur in about 26.4% of BCs in women [13]. The *PIK3CA* gene is a proto-oncogene highly mutated in many tumor types. The PIK3CA protein is activated by growth factors via direct interaction with receptors in the presence of adapter proteins such as the IRS proteins. These interactions recruit PI3K to its substrate, phosphatidyl-inositol 4,5-bisphosphate (PIP2), allowing the generation of the second lipid messenger phosphatidyl-inositol 3,4,5-triphosphate. Mutations in *PIK3CA* gene are associated with increased risk of advanced BC, resistance to hormonal treatment, increased risk of metastasis, and worse prognosis. The same mutations are correlated with being hormone receptor positive and HER2 negative in about 40% of BC patients and are associated with tumor growth, resistance to endocrine treatment, and poor overall prognosis. More recent studies have shown that the use of *PIK3CA* inhibitors has protective effects in women with advanced BC [39,40,41].

A pathogenic variant in *PIK3CA* gene was found in canine plasma following recurrent CMT and in newly diagnosed female dogs. Lee et al. [42] reported that *PIK3CA* was the most frequently mutated gene in CMT (45% of cases). Furthermore, canine *PIK3CA* A3140G (H1047R), which is known as the mutational hotspot of human BC, was also a hotspot in CMT. Targeted sequencing confirmed that 29% of CMTs had the same *PIK3CA* A3140G mutation. Integration of the transcriptome suggested that *PIK3CA* (H1047R) induces cell metabolism and cell cycle via an increase in *PCK2* and a decrease in *CDKN1B* without affecting apoptosis. The authors also identified other significantly mutated genes in the dogs, including *SCRN1* and *CLHC1*, which were not reported in the human BC. Nevertheless, we couldn’t identify *SCRN1* and *CLHC1* variants in the present study [42].

According to Sobhani et al. [43], the presence of a *PIK3CA* mutation represents an independent negative prognostic factor in BC in women. *PIK3CA* mutations in canine tumors may alter downstream molecules of PI3K/Akt/mTOR signaling pathway [44]. Alsaihati et al. [45] studied 182 samples from dogs and 886 samples from human BC, and described that CMT harbors frequent PI3K pathway alteration and *PIK3CA* H1047R mutation. They reinforced PI3K signaling as the most frequently altered pathway in both human BC and CMT.

The *BRCA2* gene also showed a large number of variants in CMT. The canine *BRCA2* is a tumor suppressor gene which encodes the BRCA2 protein, involved in DNA repair through interaction with RAD51 recombinase. This process is mediated by eight BRC repeats that are encoded by *BRCA2* exon 11. Maués et al. [46] investigated the frequency of variants in *BRCA2* exon 11 using 48 blood and tissue DNA samples from CMT. Seven single nucleotide polymorphisms (SNPs) were identified, three of which were evaluated as possibly or probably deleterious variants. Importantly, a total of 97.9% of dogs had one to three polymorphisms considering the seven SNPs identified in this study, suggesting a possible correlation between the canine *BRCA2* exon 11 polymorphisms and mammary carcinogenesis. According to a recent work by Oliveira et al. (2021) [47], SNPs are common in the *BRCA2* gene of female dogs with mammary tumors. In this work, the group studied, through liquid biopsy, germline genetic variants in 20 plasma samples from dogs with mammary cancer. Thus, eleven single nucleotide polymorphisms (SNPs) were detected, most of them in the exon 11, and two indels (deletion/insertion) in the *BRCA2* gene.

Previously, Yoshikawa et al. [48,49] studied the expression level of canine *BRCA2* gene and confirmed a reduced level in mammary tumor samples compared with healthy mammary gland, thereby associating this occurrence with canine mammary tumorigenesis. The open reading frame contained four missense variations, one insertion variation, and one silent variation, some of which were located in functional domains. Huskey et al. [50] performed whole genome sequencing on 14 purebred dogs diagnosed with mammary tumors from four breed-specific pedigrees (Golden Retriever, Siberian Husky, Dalmatian, and Standard Schnauzer) and highlighted variants in orthologs of human BC susceptibility genes. They identified variants in *BRCA2* and *STK11* genes that appear to be associated with CMT risk and demonstrated that these variants were identified in both fragment and plasma of dogs, being not frequent in women.

On the basis of liquid biopsy precision, our premise is to create clinical strategy options in oncology, with the objective of providing a more accurate and effective treatment to each patient, based on the individual genetic profile of the malignancy. Genetic diversity and dynamic change in genomic profiles of patients may be determined and accompanied by liquid biopsy, which allows for better treatment efficacy, structuring individualized therapeutic strategies [51,52].

Caveats and limitations of the study are related to the number of control samples. Because some control samples were not in sufficient quality for NGS analysis, we prioritize the evaluation of tumor and plasma samples obtained from BC patients. However, we emphasize that the number of control samples (*n* = 10) satisfies all statistical criteria by providing a reliable comparative data analysis. Moreover, among the tumor samples, there is a benign tumor, fibroadenoma. This sample remained in the study because it showed an aggressive behavior, with high expression of Ki-67 and important genetic alterations, which demonstrates the need for a more detailed study of this type of tumor.

## 5. Conclusions

We showed that liquid biopsy is useful for characterizing genetic variants and can help physicians choose a more appropriate medical intervention. In addition, liquid biopsy proved to be efficient in identifying the similarity of mutations in specific genes in both human and canine mammary tumors. Finally, liquid biopsy is an excellent method to detect new genetic mutations in the early stages and follow-up of breast cancer patients.

## Figures and Tables

**Figure 1 cancers-13-05233-f001:**
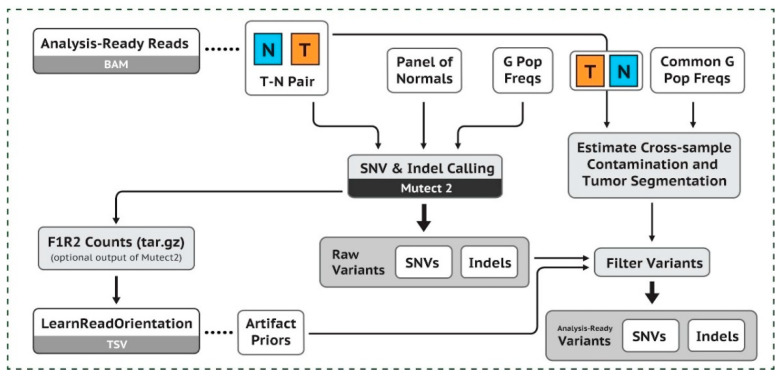
Presentation of the Workflow GAKT (https://gatk.broadinstitute.org/hc/en-us/articles/360035894731-Somatic-short-variant-discovery-SN Vs-Indels-; accessed on 20 May 2021).

**Figure 2 cancers-13-05233-f002:**
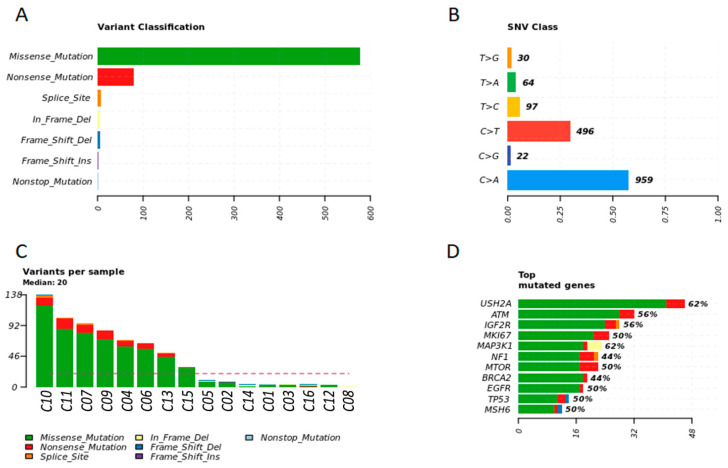
Genetic variation in tumor fragments of women with BC. (**A**) Representative variant classification. (**B**) Single-nucleotide variant (SNV) classification. (**C**) Representative variants per fragment sample. (**D**) Top mutated genes showing multiple variant mutations. Graphics were generated in the R language, using the Maftools package.

**Figure 3 cancers-13-05233-f003:**
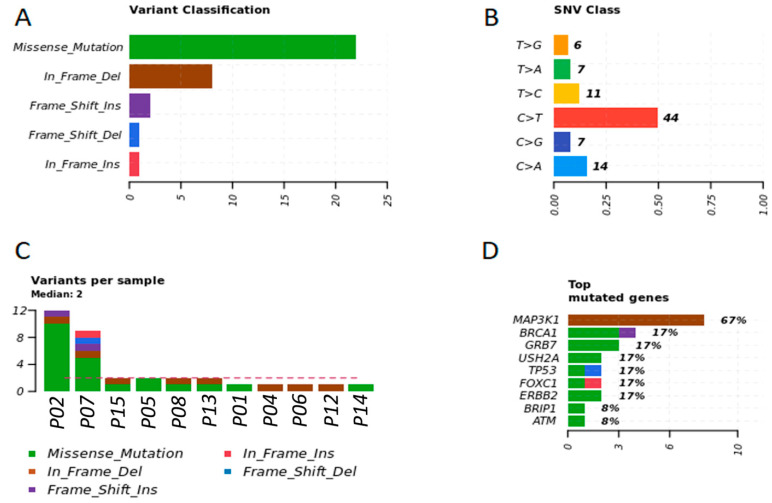
Genetic variation in plasma samples of women with BC. (**A**) Representative variant classification. (**B**) Single-nucleotide variant (SNV) classification. (**C**) Representative variants per plasma sample. (**D**) Top mutated genes showing multiple variant mutations. Graphics were generated in the R language, using the Maftools package.

**Figure 4 cancers-13-05233-f004:**
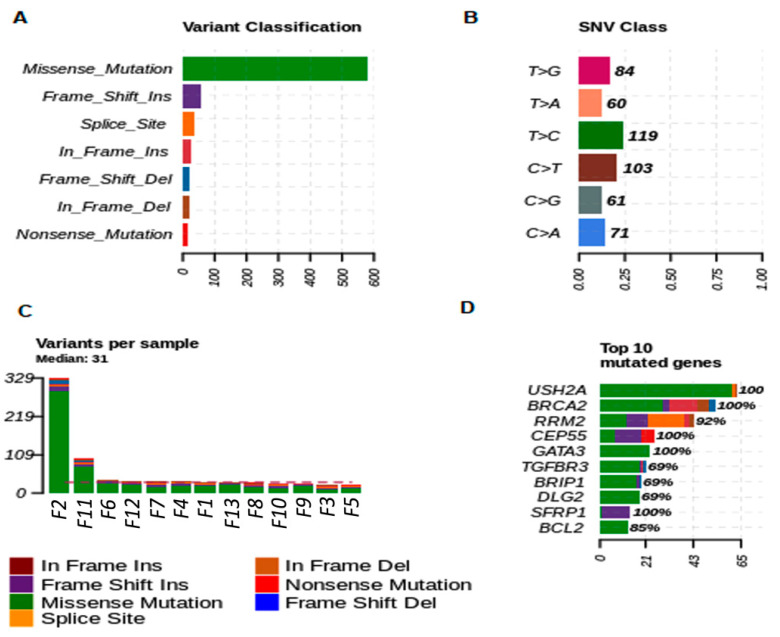
Genetic variation in tumor fragments of dogs with BC. (**A**) Representative variant classification. (**B**) Single-nucleotide variant (SNV) classification. (**C**) Representative variants per fragment sample. (**D**) Top mutated genes showing multiple variant mutations. Graphics were generated in the R language, using the Maftools package.

**Figure 5 cancers-13-05233-f005:**
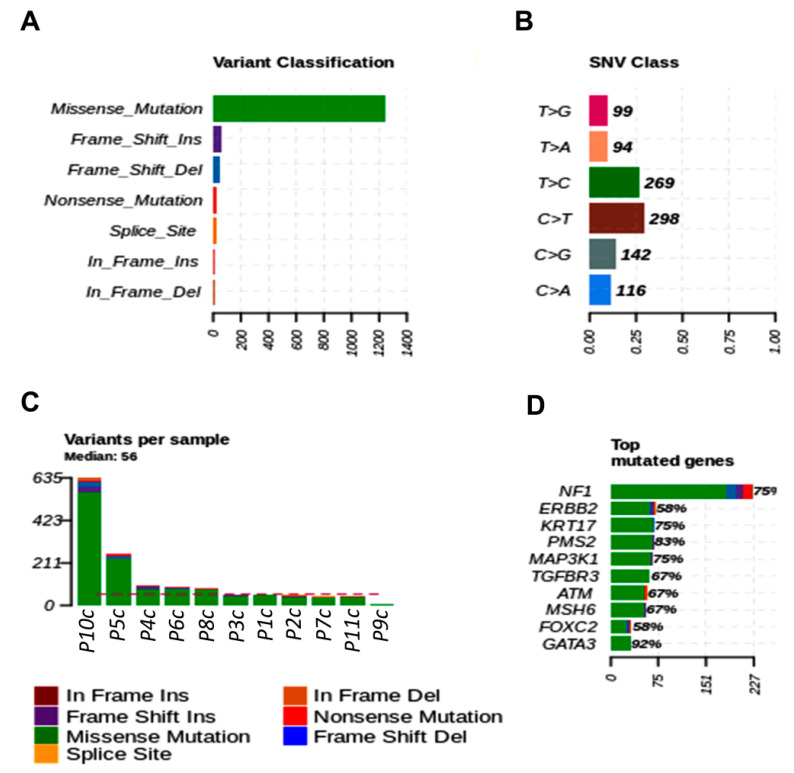
Genetic variation in plasma samples of dogs with BC. (**A**) Representative variant classification. (**B**) Single-nucleotide variant (SNV) classification. (**C**) Representative variants per plasma sample. (**D**) Top mutated genes showing multiple variant mutations. Graphics were generated in the R language, using the Maftools package.

**Figure 6 cancers-13-05233-f006:**
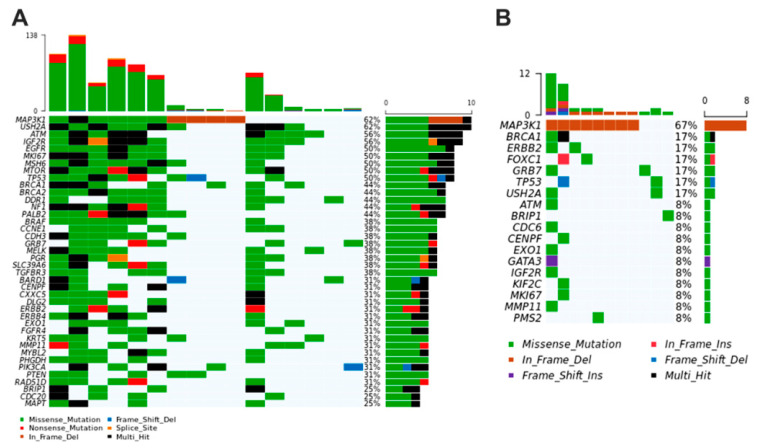
Oncoplots depicting the distribution of most representative variants in BC-associated genes regarding the individual tumor fragments (**A**) and plasma samples (**B**) of women. The upper plot shows the frequency of mutation for each tumor sample and lower left plots exhibit the mutations in each tumor sample (most deleterious mutation types are shown). The lower right plots indicate the frequency of samples mutated in fragments and plasma of patients. Graphics were generated from the R language, using the Maftools package.

**Figure 7 cancers-13-05233-f007:**
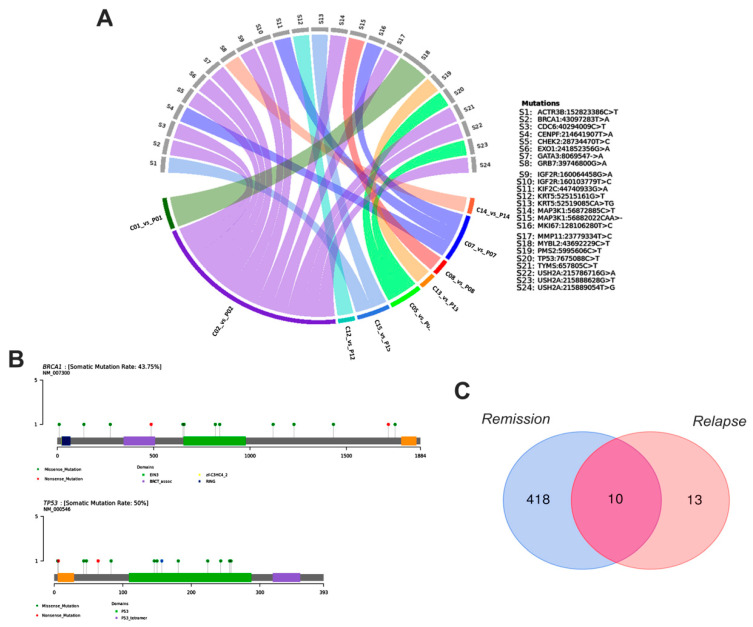
Gene variants shared by tumor fragments and plasma samples of women with BC. (**A**) Circos plot depicting the correlation between fragment and plasma shared samples and gene variants in patients. (**B**) Gene map of the somatic mutation in *BRCA* and *TP53* genes. (**C**) Total variants detected during remission and relapse of the disease. Graphics were generated from the R language, using the VennDiagram, circlize, dplyr and reshape2 packages.

**Figure 8 cancers-13-05233-f008:**
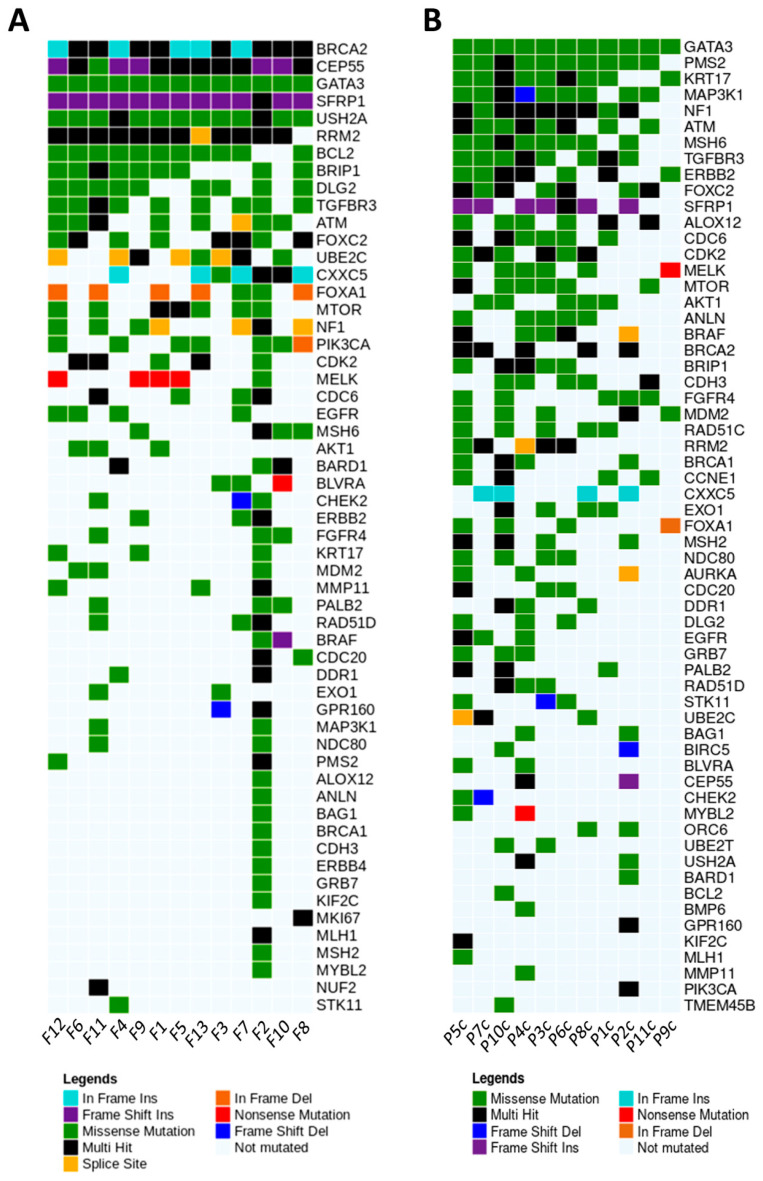
Oncoplots depicting the distribution of most representative variants in BC-associated mutated genes regarding the individual tumor fragments (**A**) and plasma samples (**B**) of dogs. The plots exhibit the mutations in each tumor sample (most deleterious mutation types are shown). Graphics were generated from the R language, using the Maftools package.

**Figure 9 cancers-13-05233-f009:**
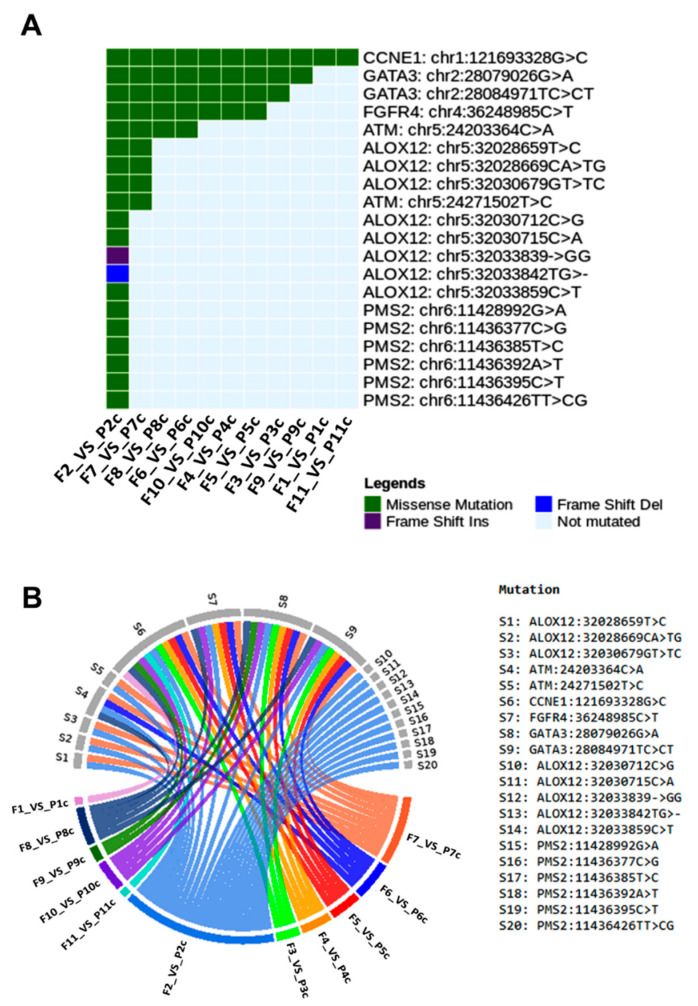
Genetic mutations shared by tumor fragments and plasma samples of dogs. (**A**) Common genetic variants in 11 corresponding samples. (**B**) Circos plot depicting the correlation between samples and gene variants in female dogs. Graphics were generated from the R language, using the Maftools, circlize, dplyr and reshape2 packages.

**Figure 10 cancers-13-05233-f010:**
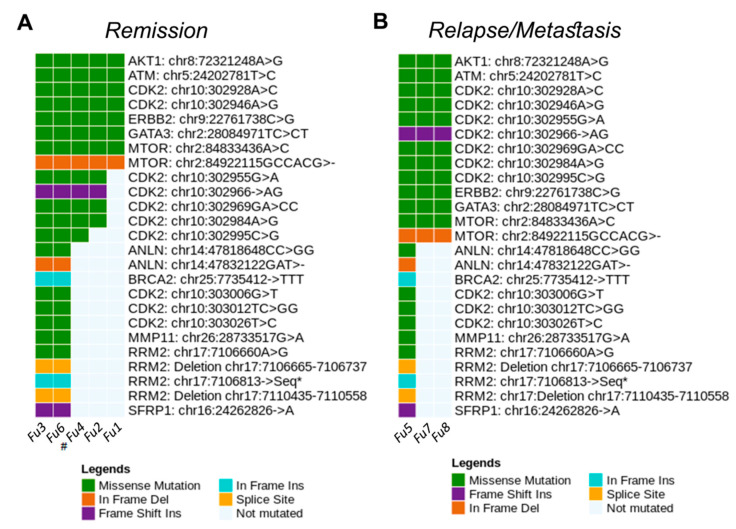
Genetic variant mutations identified in the plasma samples of dogs obtained during remission (**A**) and after disease relapse (**B**). Graphics were generated in the R language, using the Maftools package.

**Table 1 cancers-13-05233-t001:** Histopathological features of BC samples in newly diagnosed patients.

	Core				Molecular	
Plasma	Biopsy	Age	Pathology	TNM	Subtype	Immunohistochemistry
P01	C01	60	Invasive ductal carcinoma	T2N0M0	Luminal B	C.D.I Grade 2 (ER+/PR+/HER2−/Ki-67+ (20–25%)/E-cadherin+/Cytokeratin 5/6)−
P02	C02	57	Invasive ductal carcinoma	TNM1	Luminal Hybrid	C.D.I. Grade 2 (ER+/PR−/HER2+/Ki-67+ (40%)/E- cadherin+/Cytokeratin 5/6−)
P03	C03	42	Invasive ductal carcinoma	T2N0M0	Luminal Hybrid	C.D.I. Grade 2 (ER+/PR+/HER2+/K-67+ (25%)/E- cadherin+/Cytokeratin 5/6−)
P04	C04	44	Invasive ductal carcinoma	T2N0M0	Luminal B	C.D.I. Grade 2 (ER+/PR+/HER− /KI-67+(90%)/E- cadherin+/Cytokeratin 5/6−)
P05	C05	43	Invasive ductal carcinoma	T4bN3M0	Triple Negative	C.D.I. Grade 2 (ER−/PR−/Her2− /Ki-67+(95%)/E- cadherin+/Cytokeratin 5/6+)
P06	C06	61	Invasive ductal carcinoma	T2N0M0	Luminal B	C.D.I. Grade 2 (ER+/PR+/HER2+/KI-67+/E- cadherin+/Cytokeratin 5/6−)
P07	C07	61	Invasive ductal carcinoma	T3N3M0	Triple Negative	C.D.I. Grade 2 (ER−/PR−/Her2−/Ki-67+(70%)/E- cadherin+/Cytokeratin 5/6+)
P08	C08	88	Invasive ductal carcinoma	T4N1MX	Triple Negative	C.D.I. Grade 2 (ER-/PR-/HER2-/KI-67+(40%)/E- cadherin+/Cytokeratin 5/6+)
P09	C09	53	Invasive ductal carcinoma	T4N1M 0-IIIB	Triple Negative	C.D.I. Grade 2 (ER−/PR−/HER2−/Ki-67+(50%)/ E-cadherin +/Cytokeratin5/6−)
P10	C10	42	Invasive ductal carcinoma	T2N0M0	Luminal B	C.D.I. Grade 2 (ER+/PR+/HER+/Ki-67+(50%)/ E-cadherin+/Cytokeratin 5/6+)
P11	C11	62	Invasive ductal carcinoma	T4N3 M0-IIIC	Luminal B	C.D.I. Grade 2 (ER+/PR+ /HER2−Ki-67+(40%)/E- cadherin+/Cytokeratin 5/6−)
P12	C12	46	Fibroadenoma	Not applicable	Not applicable	C.DI. Grade 2 ER+/Ki67+ (50% stroma and 50% epithelium)
P13	C13	68	Invasive ductal carcinoma	T2N0M0	Luminal A	C.D.I. Grade 1 (ER+/PR+/Her2−/Ki- 67+(<10%)/E- cadherin+/Cytokeratin 5/6−)
P14	C14	90	Invasive ductal carcinoma	T3N1M0	Luminal A	C.D.I. Grade 1 (RE+/RP+/HER2−/Ki-67+(10%)/ E-cadherin+/Cytokeratin 5/6−)	
P15	C15	51	Invasive ductal carcinoma	T3N1M0	Triple Negative	C.D.I. Grade 1 (RE−/RP−/HER2−/Ki-67+(25%)/E- cadherin+/Cytokeratin 5/6−)	
P16	C16	63	Invasive ductal carcinoma	T4BN1M0	Triple Negative	C.D.I. Grade 1 (RE−/RP−/HER2−/Ki-67+(>30%)/E- cadherin+/Cytokeratin 5/6−)	

TNM classification of malignant tumors: T—primary tumor, N—regional lymph nodes, M—distant metastasis.

**Table 2 cancers-13-05233-t002:** Histopathological and clinical data of women with BC followed-up with disease recurrence.

				Molecular	
Plasma	Age	Pathology	TNM	Subtype	Immunohistochemistry
Rec1	60	Invasive Breast carcinoma	T2N2MX	Luminal B	C.D.I. Grade 2 (ER+/PR+/HER2−/Ki-67+(40- 50%)/E- cadherin+/Cytokeratin 5/6−)
Rec2	47	Invasive ductal carcinoma	T4N0M 0	Luminal B	C.D.I. Grade 2 (ER+/PR−/HER2−/ Ki-67+(70%)/E-cadherin+/Cytokeratin 5/6−)
Rec3	51	Invasive ductal carcinoma	T3N1M0	Triple Negative	C.D.I. Grade 1 (RE−/RP−/HER2−/Ki-67+(25%)/E- cadherin+/Cytokeratin 5/6−)
Rec4	51	Metastatic ductal breast cancer	T2N1Mx	HER2 positive	C.D.I. Grade 2 (ER+/PR+/HER2+/Ki-67+(40- 50%)/E- cadherin+/Cytokeratin 5/6−)

TNM classification of malignant tumors: T—primary tumor, N—regional lymph nodes, M—distant metastasis.

**Table 3 cancers-13-05233-t003:** Data from women with BC in remission.

Plasma	Age	Remission	Treatment
Rem1	83	6 years +	Chemotherapy + Radiotherapy
Rem2	58	5 years +	Chemotherapy + Radiotherapy
Rem3	55	5 years +	Chemotherapy
Rem4	46	5 years +	Mastectomy
Rem5	47	5 years +	Chemotherapy
Rem6	47	5 years +	Hormone therapy
Rem7	65	3 years +	Radiotherapy + Hormone therapy
Rem8	63	5 years +	Radiotherapy
Rem9	53	4 months +	Chemotherapy + Radiotherapy
Rem10	74	5 years +	Chemotherapy + Radiotherapy + Mastectomy
Rem11	79	3 years +	Chemotherapy

**Table 4 cancers-13-05233-t004:** Data of women used as a control group.

Plasma	Age	Information
Control1	63	Healthy women without a family history of breast cancer and without any other disease
Control2	78	
Control3	41	
Control4	70	
Control5	56	
Control6	73	
Control7	70	
Control8	62	
Control9	32	
Control10	50	

**Table 5 cancers-13-05233-t005:** Histopathological and clinical data of female dogs with newly diagnosed BC.

	Plasma	Age (years)	TNM	Pathology	Castrated	Breed
F1	P1c	10	T2N0M0	Mixed Carcinoma, grade I	Yes	MBD
F2	P2c	15	T2N0M1	Osteosarcoma	Yes	Pekinese
F3	P3c	7	T2N0M0	Complex carcinoma, grade II	No	MBD
F4	P4c	10	T1N0M0	Complex adenoma	No	MBD
F5	P5c	10	T1N0M0	Osteoma	No	Poodle
F6	P6c	7	T3N0M0	Mast cell tumor	Yes	Beagle
F7	P7c	12	T3N0M0	Ductal apocrine carcinoma	No	Dachshund
F8	P8c	16	T2N0M0	Mixed carcinoma, grade I	Yes	MBD
F9	P9c	13	T1N1M0	Tubular carcinoma, grade I	Yes	Yorkshire
F10	P10c	16	T3N0M0	Mixed carcinoma, grade I	Yes	MBD
F11	P11c	13	T2N0M0	Complex carcinoma, grade II	Yes	Belgian Shepherd

TNM classification of malignant tumors: T—primary tumor, N—regional lymph nodes, M—distant metastasis. MBD = mixed-breed dogs.

**Table 6 cancers-13-05233-t006:** Histopathological data of follow-up female dogs with BC.

Plasma	Age (years)	TNM	Pathology	Castrated	Breed
Fu1	13	*	*	Yes	Poodle
Fu2	12	*	*	No	Maltese
Fu3	10	T2N0M1	Papillary carcinoma, grade II	Yes	Labrador
Fu4	10	T2N1M1	Tubular carcinoma, grade III	Yes	SRD
Fu5	8	T2N1M0	Carcinosarcoma	Yes	Beagle
Fu6	9	T3N1M0	Mixed carcinoma, grade II	Yes	Poodle
Fu7	13	T3N0M1	Osteosarcoma	Yes	Cocker sp
Fu8	10	T2N2M1	Complex Carcinoma, grade I	No	MDB

* Information not available. TNM classification of malignant tumors: T—primary tumor, N—regional lymph nodes, M—distant metastasis. MBD = mixed-breed dogs.

## Data Availability

NGS data will be available by request from the corresponding author.

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
