# Peer review of "Liquid Biopsy as a Diagnostic and Prognostic Tool for Women and Female Dogs with Breast Cancer"

_cancers, 2021, doi:10.3390/cancers13205233_

Round 1
Reviewer 1 Report
General:
- The added value of the dog studies is not clear and should be completely omitted in my opinion. If it should be a method paper, a basic journal might be better suited. In this case, more attention should be paid to the commonalities between humans and dogs.
- The results section is a bit confused. Here it would be useful to first present a basic correlation between Primarius & liquid biopsy and then liquid biopsy / clinical outcome. This should be done on defined subgroups and not case studies.
Abstract: unstructured as now subheadings are given. It´s difficult to get the aim of this study. Also the conclusion can´t be made from the presented results
Introduction: Good, gives relevant background information
Methods: - a figure for includes patients would be helpful (consort)
- why is the controll group much smaller? are the aged-matched? should be easy to include the same numer as in the experimental arm.
Results: - whats the rationale to give number of plasma and core biopsy numbers in the tables?
- patients with missing data sould be excluded (P29)
Discussion: - very long and hard to follow, should be foccused on the essentials and not dissussing "case reports"
- it is hard or even not possible to understand the conclusion from the presented data
Author Response
Dear reviewer:
The answers to questions are attached.
Sincerely
Dr. Debora Zuccari

Reviewer 2 Report
The paper titled "Liquid Biopsy as a Diagnostic and Prognostic Tool for Women and Female Dogs with Breast Cancer" aims to attest potential and the effectiveness of liquid biopsy in the diagnosis and follow-up of women and female dogs with breast cancer (BC). The paper is well written and has the merit of publication.
There are some issues:
-How can be translated to clinical setting? Orthologues?
- How do you compare with published data?
- A resume integrative picture could be done to provide an illustration of data and transpose it.
Author Response

(The authors gave the same response as above.)

Round 2
Reviewer 1 Report
Thank you for the revised version of the manuscript in which some points have been satisfactorily changed. Even though I understand your justifications in some other points, I must state that in my opinion some points of criticism remain. Further explanations of missing data are not given in the manuscript.
1.) I still think that there should be a separation of the results of dogs and humans.
2.) A suitable control group should be presented.
3.) The tables with the plasma numbers should be revised, although this seems to be difficult.
4.) P29 with a fibroadenoma should be excluded.
5.) The abstract still lacks subheadings.
Author Response
Dear reviewer:
We are sending you the answers to your questions.
We hope to have responded to the requests.
Sincerely,
Dr. Debora Zuccari

Round 3
Reviewer 1 Report
all questions were answered satisfactorily